# Olive Leaf Extract Attenuates Chlorpyrifos-Induced Neuro- and Reproductive Toxicity in Male Albino Rats

**DOI:** 10.3390/life12101500

**Published:** 2022-09-27

**Authors:** Arwa A. Hassan, Karima Bel Hadj Salah, Esraa M. Fahmy, Doaa A. Mansour, Sally A. M. Mohamed, Asmaa A. Abdallah, Mada F. Ashkan, Kamlah Ali Majrashi, Sahar J. Melebary, El-Sayed A. El-Sheikh, Nashwa El-Shaer

**Affiliations:** 1Pharmacology & Toxicology Department, Faculty of Pharmacy & Pharmaceutical Industries, Sinai University, El-Arish 45518, Egypt; 2Biological Sciences Department, College of Science & Arts, King Abdulaziz University, Rabigh 21911, Saudi Arabia; 3Laboratory of Transmissible Diseases and Biologically Active Substances, Faculty of Pharmacy, University of Monastir, Monastir 5019, Tunisia; 4Pharmacology Department, Faculty of Veterinary Medicine, Zagazig University, Zagazig 44519, Egypt; 5Biochemistry and Chemistry of Nutrition, Faculty of Veterinary Medicine, University of Sadat City, Sadat City 32897, Egypt; 6Histology and Cytology Department, Faculty of Veterinary Medicine, Zagazig University, Zagazig 44519, Egypt; 7Theriogenology Department, Faculty of Veterinary Medicine, Zagazig University, Zagazig 44519, Egypt; 8Department of Biology, College of Science, University of Jeddah, Jeddah 21493, Saudi Arabia; 9Department of Plant Protection, Faculty of Agriculture, Zagazig University, Zagazig 44511, Egypt

**Keywords:** chlorpyrifos, olive leaf extract, neurotoxicity, reproductive toxicity, oxidative stress, apoptosis

## Abstract

Chlorpyrifos (CPF) is a common organophosphorus insecticide. It is associated with negative consequences such as neurotoxicity and reproductive injury. This study aimed to observe the ability of olive leaf extract to attenuate chlorpyrifos toxicity, which induced neuro- and reproductive toxicity in male albino rats. Olive leaf extract (OLE) exhibits potent antioxidant and antiapoptotic properties. Twenty-two mature male rats were divided into four groups: control (saline), CPF (9 mg/kg), OLE (150 mg/kg), and CPF + OLE. Treatment was administered orally for 80 days. The CPF significantly reduced serum sex hormones, sperm counts and motility, high oxidants (MDA), and depleted antioxidants (GSH, SOD, TAC) in the brain and testes homogenate; additionally, it decreased serum AChE and brain neurotransmitters, increased *Bax*, decreased *Bcl-2*, and boosted caspase-3 immune expression in neural and testicular cells. Immunological expression of Ki 67 in the cerebrum, cerebellum, choroid plexus, and hippocampus was reduced, and α-SMA in testicular tissue also decreased. Histopathological findings were consistent with the above impacts. OLE co-administration significantly normalized all these abnormalities. OLE showed significant protection against neural and reproductive damage caused by CPF.

## 1. Introduction

Organophosphate insecticides (OPs) are commonly used in agriculture to control insect pests [1]. Increased production and widespread use of pesticides harm the environment, posing immediate risks to all species, including humans [2].

Chlorpyrifos (CPF), a lipophilic organophosphorus insecticide, is commonly utilized to control foliar and subsurface crop insects [3]. The pesticide is an acetylcholinesterase (AChE) inhibitor that causes acetylcholine accumulation and neurochemical disruption, leading to death after severe exposure [4]. Many neurological deficits, such as altered blood–brain barrier integrity, attention deficit hyperactivity disorder, Parkinsonism, and dementia, are linked to consuming even small amounts of CPF [5].

CPF exhibits toxicity to multiple organs/systems, such as testes [6], reproductive systems [7], spermatogenesis, testosterone levels, DNA damage in sperm [8], hepatotoxicity, immunotoxicity, carcinogenicity, and neurotoxicity [9]. Furthermore, mechanisms of toxicity include the production of free radicals and proinflammatory cytokines, such as tumor necrosis factor-alpha (TNF-), interleukin-1 beta (IL-1), and apoptosis activation [10].

Olive leaves have been used in conventional medicine throughout Europe and countries in the Mediterranean for centuries [11]. Many flavonoid and polyphenolic chemicals are found in olive leaves; oleuropein is the most abundant molecule, accounting for 86.9% of all components. This substance has antioxidative stress and anti-inflammatory properties [12]. Oral treatment with OLE reduced oxidative stress in the midbrain by decreasing free-radical generation and increasing SOD, CAT, and GPx activities [13]. Neuroprotective properties against neurodegenerative disorders and oxidative stress-induced neurotoxicity, as well as stroke and brain damage prevention, were also observed [14]. Oleuropein is a neuroprotectant in mice’s focal cerebral ischemia/reperfusion damage [15]. 

Treatment with olive leaves also enhanced the quality and quantity of sperm, raising testosterone and luteinizing hormone levels in males. The latter activates testicular cells to produce testosterone [16]. Additionally, treatment with OLE increased total antioxidant capacity (TAC), sperm parameters, and testis antioxidant status in rat testicular tissue exposed to rotenone [17].

This research assessed the possible advantages of using OLE to counter CPF-induced neural and reproductive toxicity by evaluating serum gonadotropins, sex hormones, sperm status, oxidative status, acetylcholinesterase, brain neurotransmitters, immunohistochemistry, and histological alterations in the brain and testes of male albino rats.

## 2. Material and Methods

### 2.1. Tested Materials

Chlorpyrifos (diethyl 3,5,6-tricholoro-2-pyridyl phosphonothioate, formulation EC48%) was provided. Olive leaves were harvested, dried and powdered before extraction.

### 2.2. Extraction of Plant Material

Fresh olive leaves were left to dry in the sunlight and ground manually to a powder. Fifteen grams of olive leaf powder in a Soxhlet thimble was extracted for 4 h at 60 °C in 300 mL of 80% ethanol and 20% acetonitrile. The extract was filtered after cooling to room temperature. The filtrate solvent was removed under vacuum in a rotating evaporator at room temperature. Until used, the extract concentrate was refrigerated at 2–8 °C [18].

### 2.3. Animals

Thirty-two mature male albino rats weighing 170–200 g were kept in polypropylene cages under strict conditions of −22 °C and a 12 h/12 h light/dark cycle. The methodology followed guidelines set by the Committee of Research Ethics for Laboratory Animal Care at Zagazig University, Faculty of Veterinary Medicine, Zagazig, Egypt; (approval no. ZU-IACUC/2/F/35/2021).

### 2.4. Experimental Design

Rats were randomly divided into four groups, each with eight animals. Control animals were given distilled water in Group I; positive control animals were given CPF at a dose of 9 mg/kg bw in Group II, negative control animals were given OLE at a dose of 150 mg/kg bw in Group III, and test animals were given CPF and OLE at the above doses in Group IV.

Rats received CPF or OLE or their combination daily for 80 days. Rats had free access to food and water. Rats’ body weights were assessed weekly, and any signs of illness were recorded. Animals were euthanized by decapitation under anesthesia 24 h following the last treatment, and blood was collected for further investigation. Biochemical and hormonal tests and histological and immunohistopathological examinations were performed with brain and test samples. Semen samples were collected for semen analysis.

### 2.5. Serum Sex Hormones

Blood samples were collected and centrifuged at 5000 rpm for 10 min at 4 °C to separate serum for sex hormone estimation. Rat testosterone, luteinizing hormone (LH), and follicle-stimulating hormone (FSH) levels were estimated immediately in serum using enzyme-linked immunosorbent assay (ELISA) kits (Cusabio, Houston, TX, USA) following the manufacturer’s instructions.

### 2.6. Sperm Parameters

Sperm examination criteria (motility, viability, count, and anomalies) were examined using previously described protocols [19]. Sperm were examined under sterile conditions. The cauda epididymis was chopped in 2 mL physiological saline and incubated at 37 °C for 5 min to allow dispersion of spermatozoa. 

*Sperm motility***:** A drop of the prepared suspension was checked for spermatozoa motility with light microscopy; 200 spermatozoa were inspected in five microscope fields to record the proportion of motile sperm.

*Sperm viability***:** First, 20 µL of epididymal sperm suspension was mixed with 20 µL of eosin-nigrosin combination stain in a sterile test tube. Spermatozoa were counted randomly using light microscopy (oil immersion lens at 1000×). Sperm without a crimson or pink head was regarded as viable, and percentages of these spermatozoa were recorded [20].

*Sperm count*: A standard hemocytometer was used to determine the epididymal sperm count. Formol saline was diluted 1:4 in formen suspensions. Ten microliters of the dilution was introduced to each of five counting chambers of hemocytometers (HBG, Hamburg, Germany). Cells were settled and counted using light microscopy at 400×. The number of sperm per milliliter was reported.

*Sperm abnormalities***:** Freshly prepared alkaline methyl violet stain was to stain several semen samples films by soaking for 5 min. Films were removed from the stain and cleaned with a few drops of water before drying. In various fields, 200 spermatozoa were randomly checked for abnormalities (head or tail), and the overall anomaly percentage was calculated [21]. 

### 2.7. Oxidative Stress and Antioxidant Status Markers Determination

Malondialdehyde (MDA) is used as a marker of lipid peroxidation. A tissue homogenizer was used to prepare the brain and testicular tissues (10% *w*/*v*) in potassium phosphate buffer solution (pH 7.4) to assess MDA levels. We centrifuged homogenized samples for 15 min at 3000 rpm. 

Homogenates were treated with thiobarbituric acid using the technique described in [22]. Antioxidant enzyme activity in the brain and testicular tissues was measured as oxidant and antioxidant status indicators. Superoxide dismutase (SOD) activity was determined using nitro blue tetrazolium reduction [23]. Reduced glutathione (GSH) concentration was assessed as previously described [24]. TAC in the homogenate was evaluated following Koracevic et al. [25].

### 2.8. Serum Acetylcholine Esterase Determination

Serum acetylcholine esterase (AChE) activity was determined quantitatively using ELISA specialized kits (Cusabio, Houston, TX, USA) following the manufacturer’s protocol. Data are expressed in units of pg/mg tissue. 

### 2.9. Determination of Monoamines (Serotonin and Dopamine)

Serotonin and dopamine in brain tissue extracts were assessed with ELISA kits (Bioassay laboratory technology, Shanghai, China and ELISA Genie, Dublin, Ireland, respectively) following the manufacturer’s guidelines. Data are expressed in units of pg/mg tissue. 

### 2.10. Gene Expression

Brain and tissue samples were obtained from all rats, and 30 mg of tissue was rinsed with cold saline, snap-frozen in liquid nitrogen, and stored at −80 °C for further RT-PCR analysis. Extraction of total RNA was performed using Trizol (Invitrogen; Thermo Fisher Scientific, Inc., Waltham, MA, USA). As previously described [26,27,28], cDNA was synthesized using a HiSenScriptTM RH (-) cDNA Synthesis Kit (iNtRON Biotechnology Co., Seoul, Korea). RT-PCR was performed in a CFX96 real-time PCR detection system (Bio-Rad, Hercules, CA, USA) using TOPrealTM qPCR 2X PreMIX SYBR Green (Enzynomics, Seoul, Korea) following gene expression estimation guidelines. RT-PCR cycling conditions were as follows: initial denaturation for 15 min at 95 °C, followed by 40 cycles of denaturation for 20 s at 95 °C, annealing for 30 s at 60 °C, and extension for 30 s at 72 °C. Primer sequences used to determine *Bax* and *Bcl-2* gene expression were developed by Sangon Biotech (Beijing, China) (Table 1). The expression levels of target genes were compared to GAPDH, and relative fold changes in gene expression were estimated using the 2^−ΔΔCT^ method [29].

### 2.11. Tissue Preparation for Histopathological and Immunohistochemical Analyses

Testes were submerged in Bouin’s solution for 24 h, and the brains were fixed in a 10% neutral buffered formalin solution. Tissues were dried and encased in paraffin blocks. Under a light microscope, 5 µm thick hematoxylin and eosin (H&E)-stained sections were examined [30]. At least one deparaffinized and hydrated section from animals in each group was used to assess immune expression of caspase-3 (Catalog No. RB-1197-R7 Thermo Fisher Scientific, Waltham, MA, USA), alpha-smooth muscle actin (α-SMA) (Catalog No. ab5694, Abcam, Cambridge, UK), and Ki-67 (Catalog No. MA5-14520, Thermo Fisher Scientific, Waltham, MA, USA). As mentioned, an avidin–biotin–peroxidase technique was used [31,32]. In brief, endogenous peroxidase was inhibited by incubation for 30 min at 4 °C with 3% H_2_O_2_ in absolute methanol, and then washed with phosphate-buffered saline (PBS). A 10% normal blocking serum for 60 min at room temperature was used to prevent nonspecific reactions (Sigma-Aldrich, Cat: A9647, St. Louis, MO, USA). 

Subsequently, depending on the species, primary antibodies were incubated for 60 min with biotin-conjugated goat anti-rabbit IgG antiserum or rabbit anti-goat IgG antiserum (Histofine kit, Nichirei Corporation, Tsukiji, Tokyo, Japan). A solution of 3,3-diaminobenzidine tetrahydrochloride (DAB)–H_2_O_2_ pH 7.0 was incubated with the streptavidin–biotin complex for 3 min. After washing with distilled water, the slices were stained with Mayer’s hematoxylin counterstain. Negative control sections were generated through incubation with PBS instead of primary antibodies. A standard light microscope was used to examine stained sections, and images were captured using an AmScope (MU1403B) Digital Imaging System. ImageJ Analyzer software was used to assess the epithelial heights of 20 round seminiferous tubules from each rat and to measure the intensity of brown staining (ImageJ, NIH-Bethesda, MD, USA). Microscopic fields were selected at random and analyzed at 100× magnification.

### 2.12. Statistical Analysis

Data are expressed as means and standard deviations using the statistical package application, SPSS version 20.0 (2011, SPSS Inc., IBM, Armonk, NY, USA). Data from multiple evaluations were analyzed using one-way ANOVA and Duncan’s post hoc test. Results were deemed significant at *p* ≤ 0.05 compared to controls.

## 3. Results

### 3.1. Biochemical Parameters

#### 3.1.1. Effects of CPF Alone or Combined with OLE on Serum Sex Hormones

CPF, 9 mg/kg administered by gavage for 80 consecutive days, caused a significant decline in serum sex hormone levels. Rats treated with 150 mg/kg OLE by gavage over the same period showed nonsignificant reductions in blood testosterone, LH, and FSH versus controls. Concurrent treatment with CPF + OLE induced significant increases in serum hormone levels compared to serum from CPF-treated animals (Table 2).

#### 3.1.2. Effects of CPF Alone or in Combination with OLE on Sperm Characteristics 

Sperm count and progressive motility (%) dramatically declined in CPF-treated rats, and the numbers of immotile and malformed sperm increased compared to control and OLE-treated groups (Table 3). OLE-treated rats showed elevated sperm count and motility but no defective or unviable sperm numbers. Otherwise, sperm count and motility were significantly enhanced after OLE + CPF treatment. Compared to CPF alone, sperm viability increased significantly, and sperm malformation decreased. Thus, OLE substantially rescued the reproductive toxicity of CPF.

#### 3.1.3. The Impact of CPF Alone or Combined with OLE on Testes Oxidant/Antioxidant Markers

Oxidative damage in testicular tissues was induced by CPF exposure (Table 4). GSH, SOD, and TAC levels were significantly reduced, and MDA levels were significantly elevated. OLE-treated rats showed slightly elevated MDA, a minor depletion in GSH and TAC, and a modest increase in SOD. However, when the CPF and OLE mixture was delivered, the rats showed a strong antioxidant response with lower MDA levels and large increases in GSH, SOD, and TAC compared to CPF-rats. 

#### 3.1.4. The Impact of CPF Alone or with OLE on Brain Oxidant/Antioxidant Markers

Serum nitric oxide was significantly increased in rats orally administered CPF and insignificantly decreased in animals treated with OLE. The combination of agents significantly reduced serum nitric oxide relative to CPF-treated rats. CPF exposure induced oxidative damage to brain tissues, shown by remarkable MDA elevations and deficits in GSH, SOD, and TAC. OLE-treated rats exhibited a noteworthy reduction in MDA levels, an insignificant increase in GSH, and insignificant declines in SOD and TAC. Concurrent administration of CPF + OLE produced a marked decline in MDA and elevations in GSH, SOD, and TAC compared to CPF-treated rats (Table 5).

#### 3.1.5. The Effect of CPF Alone or in Combination with OLE on Serum Acetylcholine Esterase and Brain Neurotransmitters in Tissue Homogenates

Serum AChE and brain neurotransmitters (dopamine and serotonin) were reduced in response to CPF neurotoxicity; a slight decrease was found in OLE-treated rats (Table 6). However, (CPF + OLE) administration elicited a marked increase *(p ≤* 0.05) compared with CPF-treated animals.

#### 3.1.6. The Effect of CPF Alone or in Combination with OLE on Expression of *Bax* and *Bcl-2* mRNA in Neuronal Cells

RT-PCR showed that expression of *Bax* mRNA was dramatically upregulated and *Bcl-2* mRNA dramatically downregulated in neuronal cells in CPF-treated rats. However, *Bax* mRNA levels were significantly lowered in rats exposed to CPF + OLE, and the *Bcl-2* mRNA level was increased (Table 7).

#### 3.1.7. Effects of CPF Alone or in Combination with OLE on Expression of *Bax* and *Bcl-2* mRNA in Testicular Cells

RT-PCR revealed significant overexpression of *Bax* mRNA in testicular cells after CPF treatment, and *Bcl-2* mRNA levels were significantly downregulated (Table 8). In contrast, CPF + OLE treatment significantly reversed the overexpression of *Bax* mRNA and restored *Bcl-2* mRNA expression.

### 3.2. Histopathology and Immunohistochemistry

#### 3.2.1. Histopathological Analysis of Testes

Seminiferous tubule testes from control animals exhibited normally arranged spermatogenic epithelium and Sertoli cells. Spermatogonia combined with spermatocytes, spermatids, and spermatozoa lined the lumen. Sertoli cells rested on a basement membrane encircled by myoid cells, and a group of Leydig cells was noticed in the testicular interstitium (Figure 1a). Several seminiferous tubules in testes from CPF-exposed rats showed disorganized epithelium and loss of significant numbers of germ cells. Moreover, spermatozoa were rare or absent (Figure 1b). Hyalinization with vacuolization of the testicular interstitium was also noted. Histological structures of the testes were restored in rats treated with a combination of CPF and OLE (Figure 1c,d). Quantitative results from photomicrographs of sections from CPF-exposed rats showed lowered seminiferous epithelial height (Figure 1e). Intestinal height was increased in testes from rats treated with OLE and CPF + OLE-treated (Figure 1e).

#### 3.2.2. Immunohistochemistry of Testes

We performed immunohistochemical staining for α-SMA to evaluate variations in myoid cells. Inflammatory expression of α-SMA was observed as continuous immune positive reactions enclosing seminiferous tubules in control and OLE-treated rats (Figure 1f,h,j,l). However, only weak reactions were detected in animals treated with CPF (Figure 1g,k). Restored reactions were detected after treatment with CPF + OLE (Figure 1i,m). Statistically, the percentage area of seminiferous α-SMA immune positive expression was significantly lower in CPF-treated (1.04 ± 0.249) than in control (5.28 ± 0.474) animals. Treatment with OLE restored the expression of α-SMA (2.32 ± 0.746) (Figure 1n).

We also used immunohistochemical staining for Caspase-3 to evaluate the variations in testicular germ cell apoptosis. The immune expression of Caspase-3 was elevated in testes after CPF exposure (Figure 1o,s,p,t). In contrast, caspase-3 was reduced relative to the CPF after CPF + OLE treatment to levels similar to OLE and control animals. These findings were confirmed statistically; the percentage area of positive immune staining of seminiferous caspase-3 was significantly increased in rats exposed to CPF (40.38 ± 2.738) compared to controls (8.67 ± 0.764). Staining decreased after CPF + OLE treatment (14.74 ± 0.457). This value differed significantly from staining after CPF administration (Figure 1w).

#### 3.2.3. Histopathology of Brain Tissues

Parasagittal brain sections stained with H&E showed a nearly identical histological structure in control and OLE-treated rats (Figure 2). The cerebrum in control and OLE-exposed animals (Figure 2a,c) exhibited normal histological structure with large neuronal cell bodies and vesicular nuclei. After CPF treatment (Figure 2b), neuronal cell bodies were dark and shrunken with deeply stained pyknotic nuclei. Additionally, blood capillaries were congested. In contrast, after treatment with CPF + OLE (Figure 2d), the most abundant neuronal cell bodies appeared normal, and just a few were dark and shrunken with pyknotic nuclei (Figure 2). 

The cerebellum in both control and OLE-treated animals (Figure 2a,c) exhibited normally structured cerebellar layers, including outer molecular (OML), inner granular (IGL), and middle Purkinje cells located regularly between prior layers. CPF treatment (Figure 2b) reduced the numbers of Purkinje cell bodies with the presence of some necrotic cells and congested blood capillaries. CPF + OLE administration (Figure 2d) caused Purkinje cells to be severely jumbled. The choroid plexus exhibited a normal structure in control and OLE-treated rats (Figure 2a,c). CPF exposure (Figure 2b) caused major congestion of blood capillaries and proliferation of the epithelium lining. Combined treatment (Figure 2d) showed only congestion of blood capillaries.

The hippocampus comprises two structures: the cornu ammonis (CA) and the dentate gyrus (DG) (Figure 2a). The CA is characterized by its C-shape and is divided into four regions: CA1, CA2, CA3, and CA4. The DG is structured in a V-shape that wraps around CA4. Higher magnification of CA2 was assessed for all animals to regulate our evaluation. Control and OLE-treated rats (Figure 2a,c) showed a CA2 region comprising three layers: pyramidal cells (PL) with packed cell bodies, a polymorphic layer, and a molecular layer containing nuclei of glial cells. CPF administration (Figure 2b) produced disorganized and loosely packed cell bodies with pyknotic nuclei in the PL. CPF with OLE (Figure 2d) showed preservation of layers with few pyknotic nuclei or congested blood capillaries.

#### 3.2.4. Immunohistochemical Analysis of the Brain

We used immunohistochemical staining for Ki 67 and caspase-3 to identify changes in proliferating and apoptotic cell populations (Figure 3 and Figure 4). Immune expression of Ki-67 decreased in the cerebrum, cerebellum, choroid plexus, and hippocampus after CPF exposure. However, Ki 67 expression increased after OLE cotreatment compared to CPF alone. Ki 67 levels in OLE-treated and control animals were similar (Figure 3). Caspase-3 immunological expression increased in the cerebrum, cerebellum, choroid plexus, and hippocampus after CPF administration. OLE decreased caspase-3 expression compared with comparable levels in OLE and control rats (Figure 4).

## 4. Discussion

The excessive use of the organophosphorus pesticide CPF can induce adverse effects, such as reproductive deficits [33]. CPF exposure is also linked to nervous system disruption in both central and peripheral systems [34]. The agent inhibits AChE activity, increasing acetylcholine concentrations in synapses and causing overactivation of neurons [35]. Therefore, an urgent need exists to evaluate the neural and reproductive impacts of CPF and characterize OLE as a protective natural product to counter CPF-induced toxicity.

Chen et al. [36] found that CPF reduced sex hormone production, inhibiting spermatogenesis and inducing infertility. CPF is classified as an endocrine disruptor because of its adverse impact on reproduction via altering the pituitary–thyroid and pituitary–adrenal axes [9].

Chronic CPF exposure significantly negatively affected the adult male rat reproductive function, as evidenced by reduced sperm count, motility, and viability. Additionally, a significantly higher percentage of immotile and morphologically abnormal sperm was found after CPF exposure. These observations are consistent with previous reports [8,33,37,38,39,40]. Notably, co-administration of OLE largely restored sperm count and motility and decreased the incidence of immotile and morphologically abnormal sperm compared to CPF-intoxicated rats. Similar findings have been reported [17,41].

A decline in serum LH might be due to decreased testosterone levels. LH is involved in spermatogenesis by targeting testicular Leydig cells to inhibit testosterone synthesis [33,42]. Furthermore, active sulfur atoms, CPF metabolites, inhibit activation of cytochrome P450 3A4 (CYP3A4) [43], which participates in testosterone metabolism [44]. CPF also downregulates genes essential for gonadotropin production and steroidogenesis, which may explain its effect on FSH and LSH levels [45]. Since FSH is essential for spermatogenesis, low sperm counts after CPF treatment might be due to reduced FSH levels [42]. Concurrent administration of CPF + OLE produces a dramatic elevation in serum testosterone, LH, and FSH. Our results are consistent with [16,46]. Secretion of GnRH and testosterone is increased by oleuropein found in OLE [47]. Moreover, acidic molecules in OLE can inhibit aromatase enzyme activation, thus increasing androgens, testosterone, and dihydrotestosterone [48,49].

CPF-treated animals exhibit an oxidant/antioxidant imbalance in testis tissues, as evidenced by elevated MDA levels and decreased GSH, SOD, and TAC. The authors of [50] reported similar results. Reactive oxygen species (ROS) adversely affect the genetic content of spermatozoa and, thus, impair sperm survival and function. Furthermore, oxidative stress leads to DNA strand breaks that trigger programmed cell death [51]. Our results indicate that OLE treatment of CPF-intoxicated rats largely restored redox balance, evidenced by reduced tissue MDA and increased GSH, SOD, and TAC [16,41]. The authors of [17] reported that OLE increased TAC and decreased MDA levels in the testes of rats exposed to rotenone. High flavonoid and polyphenol content in OLE scavenges free radicals, which increases TAC concentration [52]. This content also reduces oxidative stress, upregulates SOD, and lowers MDA levels [53].

Additionally, CPF-treated rats displayed oxidative stress in brain tissues via remarkable increases in MDA concentrations and significant declines in GSH, SOD, and TAC. The authors of [4] reported similar results where CPF neurotoxicity was characterized by diminished cellular antioxidants, such as GSH, SOD, and CAT, and elevated lipid peroxidation levels. The lipophilic CPF easily penetrates the blood–brain barrier. Consequently, CPF disrupts brain integrity, promotes permeability and free-radical production, and ultimately causes CNS morphology and function alterations [54]. Changes influence the CPF-generated oxidant/antioxidant imbalance in numerous processes. For instance, antioxidant enzyme degradation is furthered by increased ROS synthesis, depletion of antioxidants, damage to DNA, and protein and lipid peroxidation [55]. 

Rats treated with OLE showed marked reversal in these adverse effects, as shown by decreases in brain tissue MDA and increases in GSH, SOD, and TAC. This OLE property is supported by [13], who demonstrated that rats treated with OLE exhibited dramatic elevation in SOD, CAT, and GPx activity and reduced MDA levels in the midbrain of intoxicated rats. These same findings were confirmed by others, suggesting that OLE is a potent antioxidant [56,57]. This property is likely associated with flavonoids and polyphenols, two key bioactive components in OLE [58]. Oleuropein efficiently suppresses MDA production in the hippocampus of an Alzheimer’s disease animal model [57]. 

AChE is critical to acetylcholine-mediated neurotransmission. This enzyme rapidly hydrolyzes acetylcholine released into cholinergic and neuromuscular synapses [59]. The current study recorded a marked decline in serum AChE and brain neurotransmitters (dopamine and serotonin) in CPF-treated rats versus untreated rats. However, supplementation with OLE significantly relieved AChE inhibition [60]. Furthermore, treatment with CPF significantly inhibited AChE in the cerebrum and cerebellum tissue in adult male rats. CPF causes cholinergic neurotoxicity by irreversibly inhibiting AChE activity, resulting in the accumulation of acetylcholine in the synaptic cleft, which overstimulates muscarinic and nicotinic receptors. Consequently, this process alters the maturation of neural pathways and synapses essential for the development of the nervous system [61]. This failure in development leads to neurobehavioral alternations and cognitive dysfunction [35]. Excess acetylcholine also suppresses anterior pituitary gland function and secondary neurotransmitter secretion, particularly dopamine and gonadotrophins [62,63,64]. Moreover, our results are consistent with previous findings that oral OLE co-administered with profenofos significantly protects against AChE inhibition [65]. 

Apoptosis is programmed cell death defined by morphological and genomic alterations in the cell. This process occurs in normal and pathological conditions, such as chemically induced cell death. CPF induces apoptosis, encouraging the production of ROS, which in turn alters mitochondrial membrane permeability. This change causes cytochrome c release into the cytosol and activates caspase-3, thus initiating an apoptosis cascade [66]. Caspase-3 is an aspartic acid cysteine protease recognized as the last enzyme in the regulation of the apoptotic process [67]. Likewise, *Bax* is recognized as a proapoptotic protein that penetrates the outer mitochondrial membrane causing the release of apoptogenic proteins into the cytoplasm. Conversely, *Bcl-2* is an antiapoptotic and antioxidant molecule located in the outer mitochondrial membrane. This protein supports cell survival and counteracts *Bax*, thus protecting mitochondrial membrane integrity [68].

The current study confirms that CPF causes apoptosis by *Bcl-2* downregulation, *Bax* upregulation, and increased immune expression of caspase-3 in neuronal and testicular cells. In contrast, OLE treatment significantly suppressed *Bax* expression and caspase-3 immuno-expression, as well as upregulated *Bcl-2* levels. Hence, OLE counters apoptosis in the brain and testes. Similarly, CPF-induced cell death in neurons and tests is associated with upregulation of *Bax* at the mRNA level, upregulating caspase-3 expression, and downregulating *Bcl-2* levels [4,69]. Conversely, OLE significantly upregulates *Bcl-2*, downregulates *Bax*, and decreases caspase 3 expression as protection against apoptosis in neural tissue [15,70]. Furthermore, OLE is an antiapoptotic agent in the testicular tissue of treated rats [16,41].

Our histological examinations revealed CPF-induced alterations in seminiferous tubules, including degenerative and necrotic alterations. The authors of [6,71] confirmed these findings. Furthermore, testes histological structures were largely rescued with co-administration of OLE. Our findings are consistent with [16,46,72]. CPF caused some remarkable histological aberrations in brain tissues. Similar observations were reported by [4,73,74,75]. However, supplementation with OLE largely restored normal histology. The authors of [13,76] observed similar phenomena.

Ki-67 is a marker of cellular proliferation [77], associated with nuclear antigens and detected in cells throughout various cell-cycle phases [78]. CPF treatment induced a decrease in Ki-67 protein-positive nuclei in the cerebrum, cerebellum, choroid plexus, and hippocampus. However, supplementation with OLE preserved neurogenic Ki-67 cell distribution. Similar results were obtained by [79,80], who reported that immunoreactive Ki-67 nuclear proteins were reduced, indicating a loss of neurogenic characteristics.

Alpha smooth muscle actin (α-SMA) is a component of the cytoskeleton and a specific marker for myoepithelial and smooth muscle cells [81,82]. It affects nearly all tissues in the body, such as the liver, lung, testis, kidney, heart, and pancreas [83].

The increased use of chlorpyrifos to control insects necessitates using a natural product such as olive leaf extract to mitigate its harmful effect on the brain and testes due to its powerful antioxidant and antiapoptotic effect.

## 5. Conclusions

The current study verified the harmful effects of CPF on brain and testes tissues, characterized by abnormalities in serum sex hormones, sperm parameters, oxidative stress, serum acetylcholinesterase, brain neurotransmitters, and apoptosis. In contrast, concurrent administration of OLE prevented the appearance of these effects, suggesting a beneficial impact against CPF neuro- and testicular toxicity in male albino rats. Further investigations are required to determine the appropriate doses of OLE in different exposure treatments and examine its effect on various organs of male albino rats such as the liver, heart, and kidney. In addition to molecular mechanisms, OLE action studies are necessary against CPF or other pesticides.

## Figures and Tables

**Figure 1 life-12-01500-f001:**
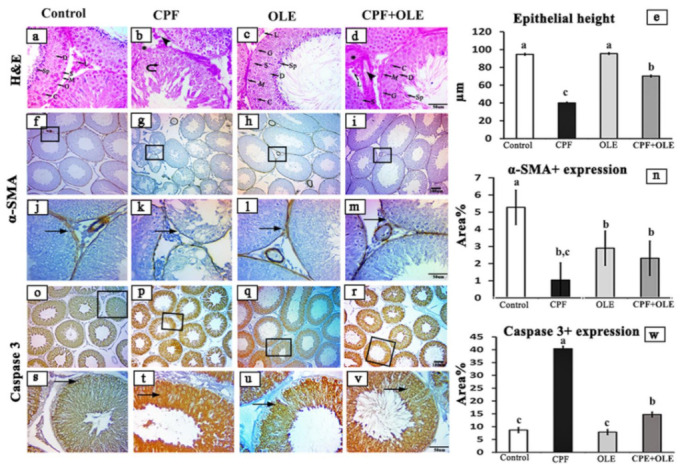
Representative photomicrographs of hematoxylin-eosin and immunohistochemical stained cross-sections of the testicular rats, in control (**a**,**f**,**o**), CPF-treated (**b**,**g**,**p**), OLE-treated (**c**,**h**,**q**), and CPF + OLE-treated groups (**d**,**i**,**r**). The testicular architecture for (**a**,**c**,**d**) groups showed seminiferous tubules lined by spermatogonia (G), spermatocytes (C), spermatids (D), immature spermatozoa (IS), and Sertoli cells (S) surrounded by myoid cells (M) and Leydig cells (L) relative to the b group, which showed disorganized spermatogenic epithelium (closed arrows) and hyalinization (asterisks) with vacuolization (arrowheads) of the testicular interstitium (scale bar: 50 μm). Expressions of Myoid cells via α-SMA (**f**–**n**) and apoptotic cell populations via caspase-3 protein (**o**–**w**) immunostaining of testicular rats, Arrows indicate dark brown staining of positive immune cells. (**j**–**m**) Moreover, (**s**–**v**) show selected areas of the seminiferous epithelium at higher magnifications. Scale bar: (**f**–**i**) and (**o**–**r**) 100 μm (**j**–**m**); (**s**–**v**) 50 μm. Bar charts demonstrating seminiferous epithelial height (**e**) and area % of α-SMA- and caspase-3-positive expressions (**n** and **w**, respectively) in the testicular sections of four experimental groups. Bars carrying different superscripts (a–c) are statistically significant differences at *p* ≤ 0.05, as determined using one-way ANOVA followed by Duncan’s test, *n* = 5/group. Values are expressed as the means ± SE.

**Figure 2 life-12-01500-f002:**
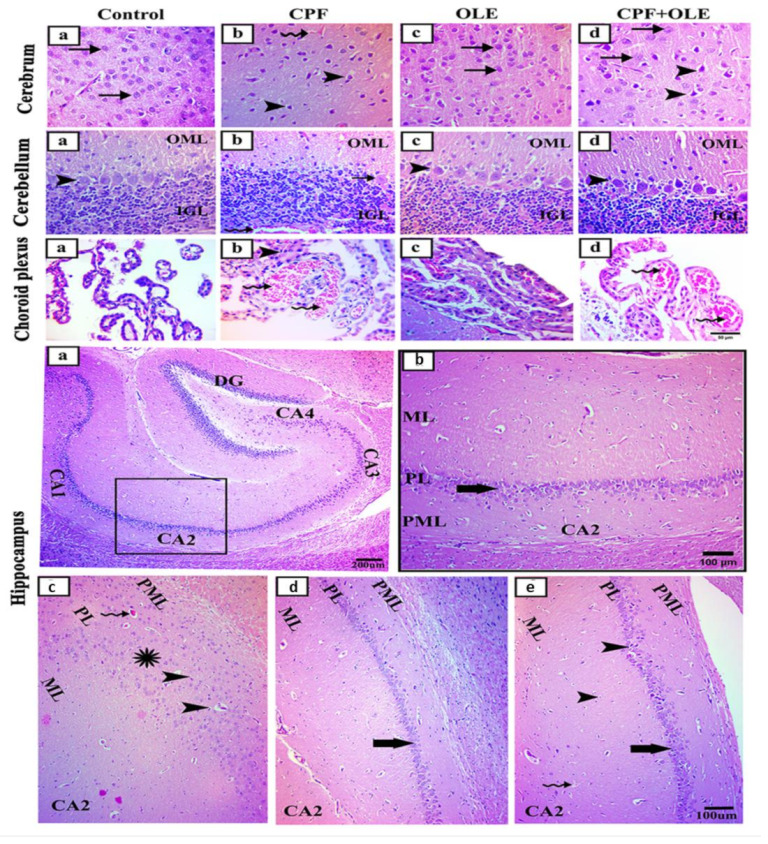
Representative photomicrographs of hematoxylin and eosin-stained parasagittal sections of the brain of rats in control (**a**), CPF-treated (**b**), OLE-treated (**c**), and CPF + OLE-treated groups (**d**). A photomicrograph of the control and OLE-treated (**a**,**b**) groups shows a normally structured cerebrum having larger neuronal cell bodies with vesicular nuclei (arrows). CPF-treated (**c**) group showing dark, shrunken neuronal cell bodies with deeply stained pyknotic nuclei (arrowheads) and congested blood capillaries (crooked arrow). CPF + OLE-treated group (**d**) showing many normal cell bodies having vesicular nuclei (arrows), while a few neuronal cell bodies appear dark and shrunken with pyknotic nuclei (arrowheads). Photomicrograph of cerebellum of control and OLE-treated (**a**,**b**) groups showing normal structured cerebellar layers as outer molecular (OML), inner granular (IGL), and middle Purkinje cells (arrowheads) that are regularly located between the OML and IGL. CPF treated (**c**) group showing a few numbers of Purkinje cell bodies with the presence of some necrotic Purkinje cells (arrow) and congested blood capillaries (crooked arrow). The CPF + OLE-treated (**d**) group showing a disorganized alignment of the Purkinje cells (arrowheads). Photomicrograph of choroid plexus: control and OLE-treated (**a**,**b**) groups showing normally structured choroid plexus. CPF-treated (**c**) group showing blood capillary congestion (zigzag arrows) and proliferation of the lining epithelial (arrowhead). CPF + OLE-treated group (**d**) showing blood capillary congestion only (zigzag arrows). The photomicrograph of the hippocampus of the control (**a**) group has the dentate gyrus (DG) and cornu ammonis (CA1,2,3,4). Higher magnification of the squared area of the CA2 region (**b**): OLE-treated (C) groups showing well-defined molecular layer (ML), pyramidal layer (PL), and polymorphic layer (PML) with PL shows closely packed cell bodies of the pyramidal neurons (thick arrow). CPF-treated (**c**) group showing an apparent disruption in CA2 PL shows disorganized and loosely packed cell bodies (asterisk) that have pyknotic nuclei (arrowheads). The CPF + OLE-treated group (**d**) showing preserved layers with few pyknotic nuclei (arrowheads) and congested blood capillaries (crooked arrow). Scale bar: (**a**–**e**) cerebrum, cerebellum, and choroid plexus, 50 μm (**a**); (**b**–**e**) the hippocampus, 200 μm and 100 μm.

**Figure 3 life-12-01500-f003:**
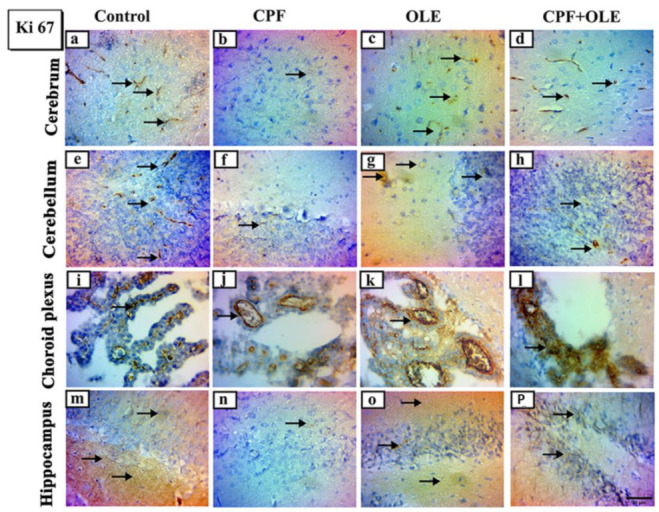
Representative photomicrographs of Ki-67 immunohistochemical stained parasagittal sections of the brain of rats in control CPF, OLE and CPF+OLE treated groups, Arrows indicating dark brown staining of immune positive proliferating cells. Photomicrograph of cerebrum of: Control and OLE treated (**a**,**c**) groups showing a moderate positive reaction. CPF treated (**b**) group showing a weak positive reaction. CPF+OLE treated group (**d**) showing a mild positive reaction. Photomicrograph of cerebellum of: Control and OLE treated (**e**,**g**) groups showing a moderate positive reaction. CPF treated (**f**) group showing a weak positive reaction. CPF+OLE treated (**h**) group showing a mild positive reaction. Photomicrograph of choroid plexus of: Control and OLE treated (**i**,**k**) groups showing a moderate positive reaction. CPF treated (**j**) group showing a weak positive reaction. CPF+OLE treated group (**l**) showing a moderate positive reaction. Photomi-crograph of the hippocampus of: Control and OLE treated (**m**,**o**) groups showing a mild positive reaction. CPF treated (**n**) group showing a weak positive reaction. CPF+OLE treated group (**p**) showing a moderate positive reaction. Scale bar; 50 μm.

**Figure 4 life-12-01500-f004:**
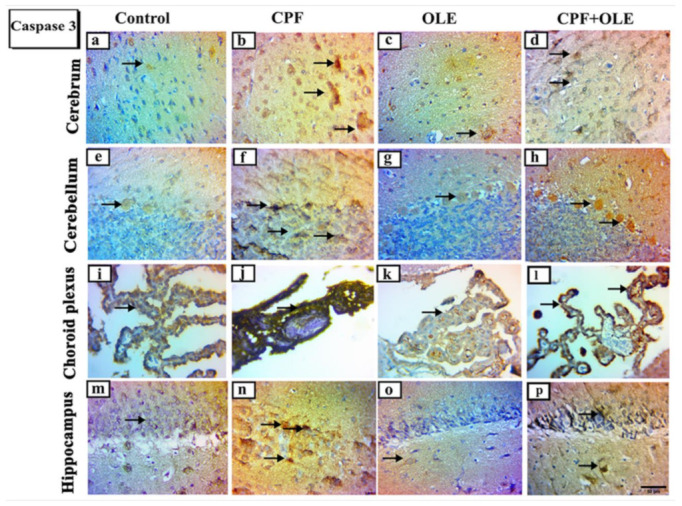
Representative photomicrographs of caspase-3 immunohistochemical stained parasagittal sections of the brain of rats in control, CPF, OLE and CPF+OLE treated groups, Arrows in-dicating dark brown staining of immune positive apoptotic cells. Photomicrograph of cerebrum of: Control and OLE treated (**a**,**c**) groups showing a weak positive reaction. CPF treated (**b**) group showing an intense positive reaction. CPF+OLE treated group (**d**) showing a mild positive reaction. Photomicrograph of cerebellum of: Control and OLE treated (**e**,**g**) groups showing a weak posi-tive reaction. CPF treated (**f**) group showing an intense positive reaction. CPF+OLE treated (**h**) group showing a mild positive reaction. Photomicrograph of choroid plexus of: Control and OLE treated (**i**,**k**) groups showing a weak positive reaction. CPF treated (**j**) group showing an intense positive reaction. CPF+OLE treated group (**l**) showing a mild positive reaction. Photomicrograph of the hippocampus of: Control and OLE treated (**m**,**o**) groups showing a weak positive reaction. CPF treated (**n**) group showing an intense positive reaction. CPF+OLE treated group (**p**) showing a mild positive reaction. Scale bar; 50 μm.

**Table 1 life-12-01500-t001:** Primers used for real-time PCR analysis.

	Forward Primer (5′–3′)	Reverse Primer (5′–3′)	Size	Accession No.
*BAX*	CGAATTGGCGATGAACTGGA	CAAACATGTCAGCTGCCACAC	109	NM_017059.2
*BCL-2*	GACTGAGTACCTGAACCGGCATC	CTGAGCAGCGTCTTCAGAGACA	135	NM_016993.1

**Table 2 life-12-01500-t002:** Effects of CPF alone or combined with OLE on serum sex hormones.

Groups	Testerone (ng/mL)	L.H (μL/mL)	FSH (μL/mL)
Control	7.703 ± 0.26 a	23.50 ± 0.34 a	3.81 ± 0.15 a
CPF	0.726 ± 0.029 c	2.05 ± 0.13 c	1.033 ± 0.11 c
OLE	7.133 ± 0.185 a	22.4 ± 0.72 a	3.53 ± 0.23 a
OLE + CPF	4.166 ± 0.176 b	11.19 ± 0.41 b	2.53 ± 0.14 b

Values are mean ± SE, *n* = 8. Values with different letters represent significance versus control at *p* ≤ 0.05.

**Table 3 life-12-01500-t003:** Effects of CPF alone or in combination with OLE on sperm characteristic.

Groups	Sperm Motility (%)	Sperm Viability (%)	Sperm Count (×10^6^)	Abnormalities	
				Primary	Secondary
Control	77 ± 1.69 b	79.30 ± 0.89 b	75.60 ± 1.49 a	4 ± 0.49 c	10.30 ± 0.39 c
CPF	0 ± 0 d	0 ± 0 d	25.62 ± 0.88 c	29.40 ± 1.09 a	42.60 ± 0.90 a
OLE	84 ± 1.24 a	85 ± 1.10 a	76.14 ± 1.93 a	3.51 ± 0.45 c	9.42 ± 0.45 c
OLE + CPF	35.12 ± 1.23 c	37.67 ± 1.40 c	48.65 ± 0.82 b	18.62 ± 1.22 b	19.93 ± 0.91 b

Values are the mean ± SE, *n* = 8. Values with different letters represent significance versus control at *p* ≤ 0.05.

**Table 4 life-12-01500-t004:** The impact of CPF alone or combined with OLE on testes oxidant/antioxidant markers.

Groups	GSH (ng/mg)	SOD (ug/mg)	TAG (ng/mg)	MDA (nmol/mg)
Control	10.76 ± 0.14 a	33.23 ± 0.39 a	21.03 ± 0.26 a	32.26 ± 0.37 c
CPF	1.60 ± 0.02 c	8.13 ± 0.18 c	4.73 ± 0.14 c	119.13 ± 0.59 a
OLE	10.70 ± 0.15 a	33.43 ± 0.23 a	20.20 ± 0.41 a	32.50 ± 0.28 c
OLE + CPF	6.10 ± 0.15 b	16.90 ± 0.37 b	11.43 ± 0.40 b	81 ± 0.57 b

Values are the mean ± SE, *n* = 8. Values with different letters represent significance versus control at *p* ≤ 0.05.

**Table 5 life-12-01500-t005:** The impact of CPF alone or combined with OLE on brain oxidant/antioxidant markers.

Groups	GSH (ng/mg)	SOD (ug/mg)	TAG (ng/mg)	MDA (nmol/mg)
Control	10.36 ± 0.08 a	32.46 ± 0.31 a	20.60 ± 0.30 a	33.53 ± 0.29 c
CPF	1.42 ± 0.04 c	8.56 ± 0.29 c	4.10 ± 0.20 c	131.16 ± 0.44 a
OLE	10.50 ± 0.28 a	31.23 ± 0.39 a	20.26 ± 0.37 a	32.20 ± 0.41 d
OLE + CPF	5.30 ± 0.15 b	21.10 ± 0.55 b	11.50 ± 0.32 b	98.26 ± 0.37 b

Values are the mean ± SE, *n* = 8. Values with different letters represent significance versus control at *p* ≤ 0.05.

**Table 6 life-12-01500-t006:** The effect of CPF alone or in combination with OLE on serum acetylcholine esterase and brain neurotransmitters in tissue homogenates.

Groups	Ach Esterase (pg/mg)	Dopamine (ng/mL)	Serotonin (ng/mL)
Control	43.50 ± 0.34 a	1.40 ± 0.05 a	1.41 ± 0.01 a
CPF	21.10 ± 0.32 c	0.24 ± 0.01 c	0.23 ± 0.01 c
OLE	43.03 ± 0.39 a	1.32 ± 0.03 a	1.40 ± 0.05 a
OLE + CPF	32.58 ± 0.37 b	0.96 ± 0.06 b	0.81 ± 0.05 b

Values are the mean ± SE, *n* = 8. Values with different letters represent significance versus control at *p* ≤ 0.05.

**Table 7 life-12-01500-t007:** The effect of CPF alone or in combination with OLE on expression of *Bax* and *Bcl-2* mRNA in neuronal cells.

Groups	*Bax*	*Bcl2*
Control	1.12 ± 0.03	1.12 ± 0.03 c
CPF	3.43 ±0.14	0.54 ± 0.05 d
OLE	0.88 ± 0.05	2.03 ± 0.18 a
OLE + CPF	2.70 ± 0.14	1.85 ± 0.14 b

Values are the mean ± SE, *n* = 8. Values with different letters represent significance versus control at *p* ≤ 0.05.

**Table 8 life-12-01500-t008:** Effects of CPF alone or in combination with OLE on expression of *Bax* and *Bcl-2* mRNA in testicular cells.

Groups	*Bax*	*Bcl2*
Control	1.24 ± 0.11 c	1.24 ± 0.05 c
CPF	4.40 ± 0.18 a	0.29 ±0.05 d
OLE	0.94 ± 0.04 d	1.72 ±0.14 a
OLE + CPF	2.05± 0.12 b	1.46 ± 0.12 b

Values are the mean ± SE, *n* = 8. Values with different letters represent significance versus control at *p* ≤ 0.05.

## Data Availability

Not applicable.

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
