# Peer review of "Olive Leaf Extract Attenuates Chlorpyrifos-Induced Neuro- and Reproductive Toxicity in Male Albino Rats"

_life, 2022, doi:10.3390/life12101500_

Round 1

Reviewer 1 Report

The topic “Olive Leaf Extract Attenuates Chlorpyrifos Induced Neuro and Reproductive Toxicity in Male Albino Rats sounds interesting and could be considered for publication, BUT there are some Minor suggestions which must be addressed before final decision.

1. Line 63, Cite (11) after centuries.

2. In order to facilitate journal readers, briefly describe how leaves are dried and powdered. 

3. I'm not sure why the portion from 93 to 98 is in italics? In line 108, B, the bio should be capitalized...

4. In Section 2.10. Ensure that the name qPCR is used consistently throughout the manuscript. Several names are used for qPCR in this section.  Additionally, gene names should be italicized throughout the manuscript including tables and figures.

5. There are many small paragraphs in the discussion. My suggestion would be to merge them. However, I will leave the decision up to the authors here.

I suggest the authors take another look throughout the manuscript, and see if some statements require more precise descriptions. The authors should find the place for their study and emphasize its importance, future aspects, and contribution to the field. It is preferable to include it in the last para of the discussion and the conclusion. The remaining writing errors are very minor, and I suppose are dealt with by the publishing office.

Author Response

Reviewer 1 comment

The topic “Olive Leaf Extract Attenuates Chlorpyrifos Induced Neuro and Reproductive Toxicity in Male Albino Rats” sounds interesting and could be considered for publication, BUT there are some Minor suggestions which must be addressed before final decision.

Response: Thanks for the reviewer for the positive comments. All comments and suggestions were considered

  1. Line 63, Cite (11) after centuries.

Response: Done as requested

  1. In order to facilitate journal readers, briefly describe how leaves are dried and powdered. 

Response: Done as requested

  1. I'm not sure why the portion from 93 to 98 is in italics? In line 108, B, the bio should be capitalized...

Response: It was adjusted accordingly

  1. In Section 2.10. Ensure that the name qPCR is used consistently throughout the manuscript. Several names are used for qPCR in this section.  Additionally, gene names should be italicized throughout the manuscript including tables and figures.

Response: It was unified accordingly

  1. There are many small paragraphs in the discussion. My suggestion would be to merge them. However, I will leave the decision up to the authors here.

Response: Thanks for the reviewer. Some paragraphs were merged

I suggest the authors take another look throughout the manuscript, and see if some statements require more precise descriptions. The authors should find the place for their study and emphasize its importance, future aspects, and contribution to the field. It is preferable to include it in the last para of the discussion and the conclusion. The remaining writing errors are very minor, and I suppose are dealt with by the publishing office.

Response: Thanks for the reviewer. The manuscript was carefully revised

Reviewer 2 Report

Dear Authors,

the paper you submitted give a new insight in potential benefical effects of olive leaf extracts in attenuating effects of organophosphorus derivatives on cellular systems. The work was clearly presented and discussed, though sometimes in a very concise form. I only suggest to revise the text for some typos.

Author Response

Reviewer 2 comments

 Comments and Suggestions for Authors

Dear Authors,

the paper you submitted give a new insight in potential benefical effects of olive leaf extracts in attenuating effects of organophosphorus derivatives on cellular systems. The work was clearly presented and discussed, though sometimes in a very concise form. I only suggest to revise the text for some typos.

Response: Thanks for the reviewer for the positive comment. The manuscript was carefully revised.